# Effects of Classic Ketogenic Diet in Children with Refractory Epilepsy: A Retrospective Cohort Study in Kingdom of Bahrain

**DOI:** 10.3390/nu14091744

**Published:** 2022-04-22

**Authors:** Simone Perna, Cinzia Ferraris, Monica Guglielmetti, Tariq A. Alalwan, Alaa M. Mahdi, Davide Guido, Anna Tagliabue

**Affiliations:** 1Department of Biology, College of Science, University of Bahrain, Sakhir Campus, Zallaq P.O. Box 32038, Bahrain; talalwan@uob.edu.bh; 2Human Nutrition and Eating Disorder Research Center, Department of Public Health, Experimental and Forensic Medicine, University of Pavia, 27100 Pavia, Italy; cinzia.ferraris@unipv.it (C.F.); monica.guglielmetti@unipv.it (M.G.); 3Salmaniya Medical Complex, Manama P.O. Box 12, Bahrain; aalaamahdi@gmail.com; 4Fondazione Policlinico Universitario Agostino Gemelli IRCCS, Largo A. Gemelli, 8, 00168 Rome, Italy; davide.guido@unipv.it

**Keywords:** ketogenic diet, pediatric epilepsy, diet therapy, seizures

## Abstract

*Background*: The classic ketogenic diet (cKD) has been used worldwide as an effective therapy for children with drug-resistant epilepsy. However, there have been no studies performed in Middle Eastern countries in order to assess the efficacy, side effects, predictors of cKD response and factors mostly associated with diet adherence. This study aims to assess the efficacy of cKD ratios of 4:1 and 3:1 and their influence on growth and biochemical parameters, particularly lipid profile and liver function tests (LFTs), and the factors most associated with diet adherence in a cohort of children with drug-resistant epilepsy in Bahrain. *Methods*: Baseline and follow-up data related to patients’ demographic and biochemical variables, epilepsy episodes, diet history and anthropometric measurements were retrieved for a total of 24 children treated with cKD in Bahrain. *Results*: After 6 months cKD initiation, 58.3% were positive responders with >50% seizure rate reduction, and 33.3% became seizure-free at 12 months. After 6 months of intervention with cKD, the level of triglycerides and albumin had a significant (*p* < 0.05) average increase over time of +1.47 mmol/L and 4.3 g/L, respectively. Although the median values of total cholesterol and alanine transaminase increased, respectively, following cKD initiation, the difference over time was not statistically significant. The mean z-scores for weight, height, and body mass index (or weight-for-length) did not change significantly at 12 months follow-up. cKD duration was the highest correlated variable with cKD efficacy (r = 0.76), which was followed by age at cKD initiation (r = 0.47). The cKD was discontinued by 14 patients (58.3%) during the first follow-up period (6 months), which was mainly due to inefficacy (*n* = 8), poor compliance (*n* = 3), food refusal (*n* = 1), achieved required efficacy (*n* = 1) and death (*n* = 1). *Conclusions*: cKD is an effective treatment for patients with drug-resistant epilepsy, and positive response to cKD was the main factor that increased adherence to the diet. Although long-term cKD could increase the risk of dyslipidemia and hepatic problems, it appears safe for children. Consequently, close monitoring and emphasis on healthy fats is of high priority.

## 1. Introduction

Epilepsy was defined conceptually in 2005 as a disorder of the brain characterized by an enduring predisposition to generate epileptic seizures. This definition is usually practically applied as having two unprovoked seizures >24 h apart. The International League Against Epilepsy (ILAE) accepted recommendations of a task force altering the practical definition for special circumstances that do not meet the two unprovoked seizures criteria [1]. The World Health Organization (WHO) estimates that around 50 million people worldwide live with epilepsy with five million people being diagnosed annually, thus making it one of the most public neurological diseases globally [2]. Epilepsy is mostly diagnosed in children and the elderly [3]. Each year, it is estimated that over 100 per 100,000 children are affected with epilepsy [4].

Epilepsy has significant economic and social impacts, including susceptibility to disability, frequent hospitalization, social isolation and discrimination, and premature death [5]. Most patients with epilepsy are successfully treated with antiepileptic drugs (AED); however, around 30% of them have drug-resistant epilepsy [6], which is defined by the International League Against Epilepsy as “failure of adequate trials of two tolerated, appropriately chosen and used AED schedules to achieve sustained seizure freedom” [7].

The classic ketogenic diet (cKD) has been used since the 1920s as an effective treatment option for many patients with drug-resistant epilepsy [8]. However, its exact antiepileptic mechanism remains unclear. The diet, which is high in fat, moderate in protein, and low in carbohydrates, results in a ketogenic state of human metabolism [9]. Nowadays, there are four main types of ketogenic dietary therapies (KDTs): classic ketogenic diet (cKD), modified Atkin diet (MAD), medium-chain triglyceride KD (MCTKD) and the low glycemic index treatment (LGIT) [10]. 

The cKD is composed of a 3:1 to 4:1 ketogenic diet ratio of fat to carbohydrate and protein [11]. The MAD is composed of a 1:1 to 2:1 ketogenic diet. The least restrictive type is the LGIT, which allows for 40 to 60 g of carbohydrates per day with a low glycemic index below 50. All these diets are similar in their efficacy in children [12,13].

A large percentage of patients with epilepsy fail to respond to the AEDs. Furthermore, uncontrolled epilepsy leads to frequent hospitalization and negatively impacts patients’ and caregivers’ quality of life. For many, KDTs represent the last hope for a lasting relief of their epilepsy [14,15].

However, several factors may interfere with the adherence to KDTs including restrictiveness of the diet, adverse side effects and lack of efficacy [16]. 

The carbohydrate restriction in KDTs is one of the challenges that made the diet adherence more difficult due to the limited food options and decreased palatability—particularly when considering the dietary patterns of the country. For example, in Bahrain, the carbohydrate foods such as rice and bread are considered a staple food and a main component in almost every meal. Furthermore, almost all of the traditional Bahraini foods are carbohydrate-based such as harees, jareesh, balaleet, qaimat, khanfaroosh and machboos.

Because of this, the adherence of the cKD is a critical factor to consider in the Middle East. The features of the ketogenic diet in the Middle East are based on a mixing contamination of local and foreign foods. Basically, it is a fusion of several food cultures including Arabic, Turkish, Iranian, Iraqi, Indian, Pakistan, Lebanon and Sub-Saharan Africa.

The ketogenic diet in the Middle East includes the use of herbs and spices with high-fat food preferences including heavy cream, cultured cream, butter and safflower oil. Protein-rich food choices include lamb, fish, chicken and beef. Olives (high in fat) are grown in these regions, and children are accustomed to eating them regularly. Very common in this area is the attitude to eat nuts such as pistachios that make great snacking options for anyone following the ketogenic diet. They are excellent sources of both dietary fat and fiber. Very common are the coconut biscuits that could provide an alternative for a fitting breakfast.

The Middle Eastern diet is very suitable for adapting itself to a ketogenic diet, since there are a variety of local recipes and foods with more than 70% of fats and less than 10% of sugar. As showed in Figure 1, traditional foods that could be part of this regime are muthabal or babaganoush (smoky and creamy eggplant with oil), hummus (mashed chickpeas with oil), Shakshuka (creamy eggs with pepper), falafel (dried chickpeas mixed parsley), beef grill kebab, chicken shawarma and lamb Shashlik Kebab. Even the most common local fish rich in fats such as Hamour and Safi are very ketogenic friendly.

The most common side effects reported were gastrointestinal disturbance, cardiovascular complaints, and renal and skeletal problems [12,13]. Growth retardation is another concern associated with cKD which has been investigated in many studies [17,18]. Patients experiencing adverse side effects are less likely to be adherent to the cKD [13]. Moreover, the lack of efficacy was also the main reported reason for cKD cessation in several studies [14,15]. Nonetheless, cKD remains an effective and non-pharmacologic option for childhood refractory epilepsy.

From this literature review, apparently, there is a lack of a study conducted on cKD in children with epilepsy in the Middle East region; consequently, the significance of performing such a study in this region is increasing. In addition, it is important to ensure the safety of cKD in terms of its impact on children’s growth and biochemical markers, especially lipid profile, which continued to be controversial. Furthermore, detecting the variables that are most likely linked with favorable response to cKD will assist in the improvement of cKD therapy protocol in children with epilepsy.

It can be hypothesized that cKD might convey risks of increased lipid profile, liver enzyme derangement, and declined growth in children with drug-resistant epilepsy. In addition, the adherence to cKD will increase with increased efficacy and tolerance.

This study aims to assess the efficacy of cKD and its influence on growth and biochemical parameters, particularly lipid profile and liver function tests (LFTs) and the factors most associated with diet adherence in a cohort of children with drug-resistant epilepsy in Bahrain. Specifically, this study was performed to identify the factors that can predict the cKD efficacy, to measure the dropout rate, to distinguish the factors that enhance ketogenic diet adherence, to investigate the association between different factors related to patients, seizures, and diet treatment and the response to ketogenic diet and to assess the influence of ketogenic diet on the growth of children with epilepsy.

## 2. Materials and Methods

### 2.1. Study Setting

This is a retrospective cohort study conducted at the Salmaniya Medical Complex (SMC) in Bahrain between March 2018 and June 2021. The study protocol was approved by the Secondary Health Care Research Subcommittee (SHCRC) of the Ministry of Health, Bahrain (serial number: 23250221; approved on 25 February 2021). The study was conducted in full compliance with the Declaration of Helsinki, and written parental consent was obtained. 

### 2.2. Patients and Data Collection Protocol

The cKD has been used as a treatment for children with drug-resistant epilepsy at SMC since 2010. Patients are monitored by a multidisciplinary team including pediatricians, neurologists and licensed dietitians. Patients were under treatment with these AEDs drugs: clobazam, oxcabazepine, levetiracetam, topiramate, phenobarbiton, clonazepam, sabril, vigabatrin, rivotrel and clonazepam.

Medical records were used retrospectively to identify all infants and children aged 0 to 14 years who were treated with cKD for drug-resistant epilepsy at SMC between 2010 and 2021. Patients’ demographic data were collected at baseline including sex, age at epilepsy onset, age at cKD initiation, epilepsy duration (time between epilepsy onset and cKD initiation), epilepsy diagnosis/etiology, number of AEDs used, and seizure frequency. Furthermore, biochemical (total cholesterol, TC; triglycerides, TG; alkaline phosphatase, ALP; alanine transaminase, ALT; gamma-glutamyl transferase, GGT; and albumin) were collected at baseline and at outpatient clinic follow-ups at 6 months, and anthropometric (weight, height, and body mass index, BMI) data were also collected at baseline and at outpatient clinic follow-ups at 12 months. 

cKD efficacy in patients was categorized into three categories: responders (>50% seizure reduction), non-responders (<50% seizure reduction) and seizure freedom after a follow-up at 3, 6 and 12 months. Diet information of cKD type, route of feeding, cKD duration and reason of cKD discontinuation were recorded as well. 

The WHO growth reference and Centers for Disease Control and Prevention (CDC) reference chart were used to determine z-scores for weight-for-age, height-for-age, BMI-for-age, and weight-for-length. WHO growth standards were used for children less than 2 years old, while CDC growth charts were used for children aged 2 years and older as recommended by the CDC. Differences in baseline and 12 months follow-up after cKD initiation z-scores were calculated for height (∆ height z-score), weight (∆ weight z-score) and BMI (∆ BMI z-score).

### 2.3. Ketogenic Diet Protocol

Following a standardized protocol [3], patients were admitted to SMC for cKD initiation for an average of 3–4 days. The macronutrients distribution of the prescribed diet was around 90%, 6% and 4% of total calories intake from fat, protein and carbohydrate, respectively. Compared to the normal diet that is typically composed of 45–65% of calories from carbohydrate, 20–35% from fat and 10–35% from protein, it may seem an unbalanced diet. The diet was advanced daily by either one-third energy interval, with a constant cKD ratio, until full energy requirements were achieved, or by daily advancement of a cKD ratio from 1:1 or 2:1 to 3:1 or 4:1 with total energy provided. The fasting protocol (12 h) prior to KD initiation was applied for all patients between 2010 and 2018. Since 2019, the non-fasting protocol was utilized following the updated recommendations of the International Ketogenic Diet Study Group on the optimal management of children receiving the cKD [3]. No supplementations of multivitamin or potassium citrate have been included into the protocol. The KetoCal^®^ KD formula (Nutricia North America, Gaithersburg, MD, USA) was used as the sole source of nutrition for both infants < 6 months old and enterally fed patients. In addition, KetoCal^®^ was used to supplement the intake of other orally fed patients. During admission, patients were closely monitored for ketosis and hypoglycemia. After verification of cKD tolerance and positive urinary or blood ketones was achieved, patients were discharged from the hospital and followed up in the dietetic outpatient clinic every 3 and 6 months to monitor cKD efficacy and tolerability and to assess growth and biochemical status. 

After discharge, caregivers were instructed to monitor urine ketones several times per week to ensure cKD adherence and monitor the ketosis level. If the level of ketosis was low at follow-up (urine ketones less than 80 mg/dL), the ketogenic ratio was increased. After 6 months of cKD, the discontinuation rate was reported. This paper did not include data about the hydroxybutyrate, since the database had incomplete data related to this outcome, since the data collection was provided by the caregivers.

### 2.4. Statistical Analysis

A retrospective open label, experimental pre–post study was designed, and sample size was not determined a priori because the subjects were *n* > 12, as previously suggested by Julious [19]. R software version 3.5.3 was used to perform the statistical analysis. Descriptive statistics of the sample were performed using median and interquartile range values and frequencies. Attrition was treated by complete cases analysis, as recommended by the Committee for Medicinal Products for Human Use for exploratory studies [20].

Non-normally distributed data were checked by Shapiro–Wilk test. Generalized estimating equation (GEE) models [20,21] for repeated measures were employed to ascertain the changes in outcomes among individuals across time. GEE models were selected due to their strength for missing values and the possibility of managing the intra-subject variability resulting from the different measurements performed on each patient [22]. For each outcome, a Gaussian GEE model was fitted in which the time reflected a categorical two-level variable (i.e., follow-up measurement (post-) or baseline (pre-), with baseline as reference) with the ‘‘focus’’ predictor. An ‘‘identity’’ link was used to fit the Gaussian GEE models. The GEE time parameters (ΔT) were interpreted as mean differences of the outcome at the follow-up time (t1) from baseline (t0). Furthermore, a stratified analysis considering the two cKD types, 4:1 (*n* = 9) and 3:1 (*n* = 13), was performed. A GEE model was fitted, where the time reflected a categorical two-level factor, based on the number of time points (i.e., baseline (t0) as reference and the last measurement (t1)) and the type (i.e., “4:1” or “3:1”, with “3:1” as reference), and their interaction terms were ‘‘focus’’ predictors. Accounting for these GEE models, ΔT were interpreted as mean differences of the outcome at the time point t1 from baseline in the 3:1 group; the interaction parameters (ΔTxTy) were interpreted as mean differences of the outcome between type at a given time point. Finally, type-associated parameters (ΔTy) were interpreted as baseline mean differences between type. Wald tests were performed, and statistical significance for model variables was assessed by 95% confidence intervals. Two-tailed *p*-values <0.05 were considered to be statistically significant.

## 3. Results

### 3.1. Demographic and Clinical Data

Initially, 37 epileptic patients treated with cKD at SMC were identified from medical records. However, 13 patients were excluded due to insufficient information. Therefore, the final study cohort consisted of 24 children (13 males and 11 females) who were treated with cKD for drug-resistant epilepsy. The demographic and clinical characteristics of the study sample are summarized in Table 1. The seizure etiologies were known only in 11 patients: genetics (*n* = 8), structural (*n* = 2) and infectious (*n* = 1). The identified epilepsy syndromes were infantile spasm (*n* = 5), epileptic spasm (*n* = 5), Ohtahara syndrome (*n* = 3), myoclonic epilepsy (*n* = 2) and one case each of Rett syndrome, multifocal seizure syndrome, complex partial seizure, myoclonic encephalopathy, Lennox–Gastaut syndrome and Aicardi syndrome. The mean age at epilepsy onset was 6.25 months (age ranged between 4 months and 11.25 years), whereas at cKD initiation, it was 27.6 months (age ranged between 4 months and 8 years). All patients received cKD, with a 3:1 ratio in 62.5% (*n* = 15) and 4:1 ratio in 37.5% (*n* = 9) of the patients, respectively. Oral feeding was the administration route for the majority of the study population (79.17%, *n* = 19), except for five patients (20.83%) who were on enteral feeding. The mean duration of the cKD was 22 months (range between 3 months and 6 years). The mode of AEDs drugs was 2 (IQR: 1–5).

### 3.2. Efficacy on Seizure Reduction at 3 and 12 Months Follow up

In total, 14 patients (58.3%) were cKD responders with >50% seizure reduction at 3 months, and one-third (33.3%) achieved seizure freedom at 12 months. However; 10 patients (41.7%) were non-responders (<50% seizure reduction).

### 3.3. General Characteristics of the Sample

As shown in Table 1, 24 children (13 males and 11 females) were treated. The seizure etiology was known in 11 patients: eight patients had genetic causes, two had structural causes, and one was infectious. The identified epilepsy syndromes were Ohtahara syndrome (*n* = 3), Rett syndrome (*n* = 1), epileptic spasm (*n* = 5), multifocal seizure syndrome (*n* = 1), complex partial seizure (*n* = 1), myoclonic encephalopathy (*n* = 1), infantile spasm (*n* = 5), Lennox–Gastaut Syndrome (*n* = 1), Aicardi syndrome (*n* = 1), and myoclonic epilepsy (*n* = 2). 

The mean age at epilepsy onset was 7 months (IRQ: 36 months), whereas at cKD initiation, it was 24 months (IQR 28.5 months). All patients received cKD, with a 3:1 ratio in 62.5% (*n* = 15) and a 4:1 ratio in 37.5% (*n* = 9) of the patients. Oral feeding was the administration route for the majority of the study sample (79.17%, *n* = 19), except for five patients (20.83%) who were on enteral feeding. The median duration of cKD was 21 months (IQR 72 months).

### 3.4. Effect on Biochemical Markers

Table 2 shows the median and IQR for biochemical markers at baseline and 6 months follow-up. All biomarkers were within the normal range. In Table 3, concerning GEE modeling, the previous significant mean increases of triglycerides (ΔT = 1.47, *p* = 0.002) and albumin (ΔT = 4.30, *p* = 0.034) were detected by pre–post modeling (see Table 3). The significant increase in triglycerides was also confirmed in the cKD type interaction modeling (ΔT = 1.46, *p* = 0.010). Although the mean value of TC and ALT increased by 0.21 mmol/L and 6.29 U/L, respectively, after cKD initiation, the difference was statistically not significant. Additionally, insignificant reductions of both ALP and GGT mean levels were observed. Notably, no cKD type interactions resulted in significant biochemical outcomes (Table 3). Figure 2 shows the interaction plots of the biochemical outcomes between time and cKD type (cKD ratio 3:1 versus cKD ratio 4:1).

### 3.5. Effects on Growth

Table 4 shows the change in growth after 12 months of cKD. From baseline to 12 months follow-up, the median z-scores for weight, height and BMI (or weight-for-length) z-score did not show a statistically significant change.

### 3.6. Predictive Factors for cKD Efficacy

Figure 3 represents the correlations between positive response to cKD (>50% seizure reduction) and the different baseline variables. The blue lines denote positive correlations, and the numbers represent the correlations between variables.

cKD duration was the highest correlated variable with cKD efficacy (0.76), which was followed by age at xKD initiation (0.47). The number of AEDs drugs was not correlated with the duration of cKD.

### 3.7. Reasons for cKD Discontinuation

The cKD was discontinued by 14 patients (58.3%) at 6 months. The reported reasons of discontinuation were inefficacy (*n* = 8), poor compliance (*n* = 3), food refusal (*n* = 1), achieved required efficacy (*n* = 1) and death (*n* = 1). The death case had lost follow-up, and death happened outside Bahrain. Although no detailed information could be obtained, it was most probably related to the disease complications and not the cKD.

## 4. Discussion

The cKD was effective in reducing seizures by over 50% in 58.3% of children after 3 months of cKD initiation in SMC in order to treat drug-resistant epilepsy. Furthermore, 33.3% of the total sample achieved seizure freedom after 1 year of cKD initiation. This study reports similar results in accordance with Chen and Kossoff [23], who reported a response rate of 55% and seizure freedom in 35% of their total sample following cKD initiation. Nevertheless, the response rate observed in this study was lower than that reported by several previous studies which ranged between 70 and 90% [24,25,26,27,28,29,30]. This might be explained by the low retention rate (41.7%) in this study. Out of a total of 24 children who started the cKD, 14 patients (58.3%) discontinued the diet at different time points for several reasons.

The present study examined the impact of cKD on the biochemical indices of the children, particularly lipid profile and LFT. In terms of lipid profile, an increase was observed in both triglycerides and TC levels after cKD treatment, but the increase was significant only for the triglycerides. These findings are in line with both Dressler et al. [31] and Pires et al. [32] who have shown an increase in triglyceride level with the cKD. Furthermore, TC levels were significantly higher in the cKD group compared to the care-as-usual (control) group as reported by Lambrechts et al. [33] in their randomized control trial of children with refractory epilepsy. Hyperlipidemia is a well-known side effect of almost all KDTs, but the effect may be transient [34] and does not prevent continuation of the treatment with adequate diet modifications [35,36,37,38]. Further studies to evaluate the long-term risk of these elevations in lipid profiles in both children actively receiving the ketogenic diet and those who discontinued it years prior are necessary.

Arslan et al. [39] investigated the hepatic effect of cKD in children with drug-resistant epilepsy and reported elevated levels of ALT and AST after one month of cKD initiation. Similarly, an increase in ALT levels was also observed in this study. On the other hand, ALP and GGT levels were reduced during the cKD. More studies are needed to explain the cKD-induced liver problems.

Hypoalbuminemia was previously reported as one of the side effects of cKD [40]. This is in contrast with the present study where a significant increase in albumin was observed during the cKD period. On the other hand, Martins et al. [41] and Arslan et al. [39] both reported insignificant changes in albumin levels with cKD. This improvement in albumin levels in the study sample can be attributed to the adequate protein in the cKD meal plans provided to the patients containing high biological value protein sources. In addition, this study found no significant variation in biochemical markers in terms of different cKD ratios (3:1 versus 4:1). This finding is in line with a previous randomized open-label study which found no significant difference in biochemical parameters between different KD ratios (4:1 versus 2.5:1) [41].

When considering the growth of children who started the cKD, the mean z-scores for height, weight, and BMI (or weight-for-length) did not change significantly at 12 months follow-up compared to the baseline values. The present finding is comparable to the findings of Tagliabue et al. [42] and Ferraris et al. [17] who reported constant growth z-scores after 6 and 12 months of cKD, respectively. 

These results can be explained by the fact that patients of this study had received adequate calories, in addition to the intensive follow-up program provided at SMC, which allowed patients access to dietitians by phone for diet troubleshooting and ensuring compliance with the KD that meets energy requirements. Furthermore, most of the study patients were receiving a specialized ketogenic formula to supplement their food intake, which further improved their calorie intake. In addition, more importantly, this result indicates that calorie needs can be fulfilled by cKD to promote the normal growth of children.

The findings of this study found a favorable response to cKD being correlated with the duration of cKD, age at cKD initiation, epilepsy duration, and number of AEDs at baseline. These factors have also been identified in several studies as predictors of cKD efficacy [24,32,43]. However, in line with Coppola et al. [44], cKD efficacy in this study cohort was not associated with sex and baseline weight. Interestingly, cKD duration was the most variable positively correlated with efficacy, which suggests that patients are more likely to adhere to the cKD if they achieved a positive response.

In the present study, inefficacy was the most reported reason for discontinuing the cKD by the majority of the study sample followed by poor compliance. These results coincide with previous studies which reported that inefficacy was the factor most associated with cKD withdrawal [14,15,22,23,24,25,37,41,42,43,44,45]. Acceptance of the diet was another factor that contributed to the compliance and adherence of the diet. This is in line with the Indian study by Baby [46], who reported that poor compliance was one of the most common reasons of KD discontinuation in 74 south Indian children whose usual diet is rich in carbohydrates, especially rice. Thus, the cultural tolerability was a main contributor to cKD adherence, which emphasizes the importance of customizing recipes to better suit the cultural dietary preferences. A less restrictive diet (LGIT) was found to be more acceptable than the more restrictive cKD in an Iranian study by Karimzadeh et al. [47]. This suggests that these less restrictive diets can be used as another approach to increase cKD adherence.

Despite the efficacy of cKD in the treatment of drug-resistant epilepsy, it may convey clinical and nutritional risks, including hyperlipidemia and hepatic side effects. In this study, a significant increase in triglycerides was observed after cKD initiation, as well as the increase in levels of the liver enzyme ALT. Consequently, it is highly recommended to closely monitor liver enzymes and lipid profile and to focus on healthy fats such as monounsaturated and polyunsaturated fats instead of unhealthy saturated fats when preparing dietary plans for these patients. However, the cKD had no significant influence on the growth parameters of the studied patients at 12 months follow-up.

cKD efficacy was highly correlated with the duration of cKD in this study, which indicates that a positive response increases the adherence to cKD. Furthermore, cKD tolerance in terms of absence of side effects is another factor that affected the adherence in the cohort.

Interestingly, this study is the first study conducted in Bahrain to evaluate cKD treatment for childhood drug-resistant epilepsy, which makes it a valuable contribution in the existing literature. However, like other studies, this study has a number of limitations, including its retrospective nature and the small sample size. In addition, it was not possible to exclude the effect of AEDs on liver enzymes. Moreover, the absence of ketonemia levels was another important limit.

It is highly recommended that future research includes larger sample sizes and a prospective study design to assess the nutritional and clinical effects of cKD, both in the short and long term. In addition, other benefits of cKD in terms of quality of life can be investigated in patients with epilepsy. Finally, other less restrictive types of cKD including MCTKD, modified Atkins and low-glycemic index diets could be evaluated in future studies.

## Figures and Tables

**Figure 1 nutrients-14-01744-f001:**
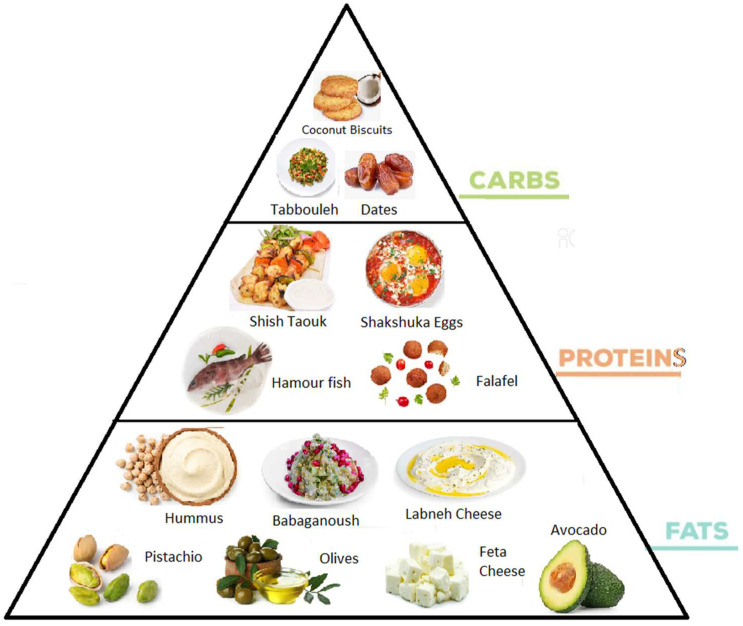
Ketogenic diet foods in the Middle East.

**Figure 2 nutrients-14-01744-f002:**
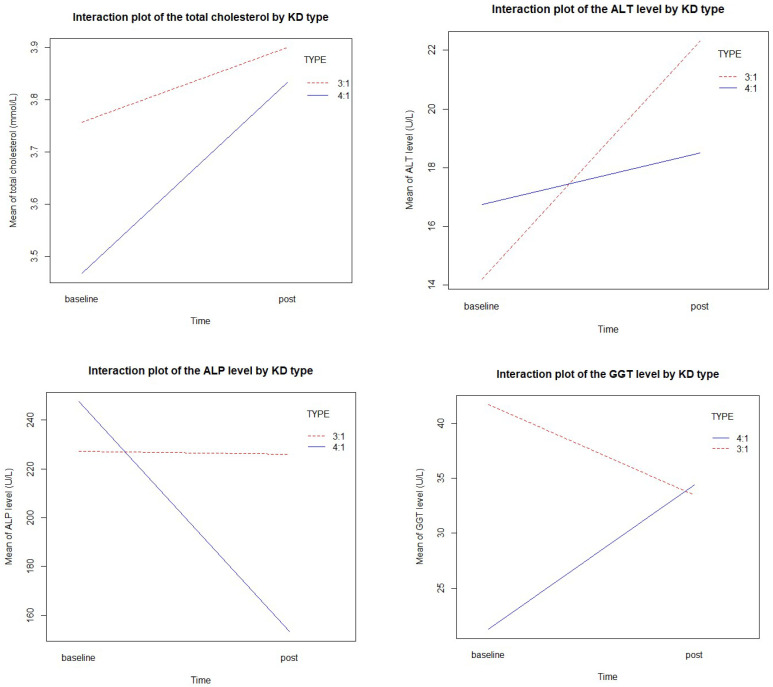
Interaction plots of the biochemical outcomes between time and cKD type.

**Figure 3 nutrients-14-01744-f003:**
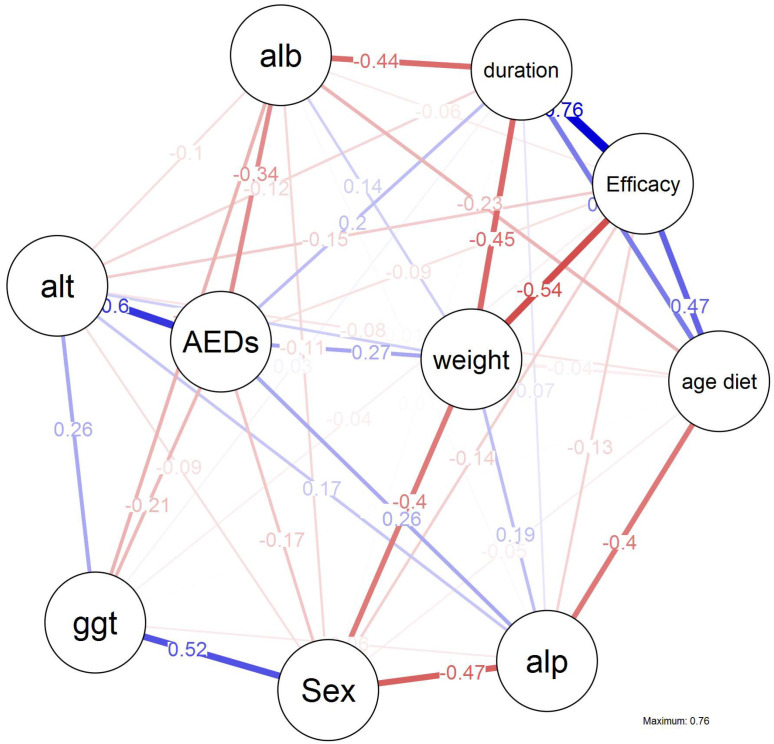
Partial weighted correlations network analysis between ketogenic diet efficacy and different variables.

**Table 1 nutrients-14-01744-t001:** Demographic, clinical characteristics and drop-out rate.

	*n* (%)
Gender	11 Females (45.83)
	13 Males (54.17)
Seizure etiology	Genetic, 8 (33.33)
	Structural, 2 (8.33)
	Infectious, 1 (4.17)
	Unknown, 13 (54.17)
Epilepsy syndrome	Epileptic spasm, 5 (20.83)
	Infantile spasm, 5 (12.5)
	Ohtahara syndrome, 3 (12.5)
	Myoclonic epilepsy, 2 (8.33)
	Lennox–Gastaut syndrome, 1 (4.17)
	Multifocal seizure syndrome, 1 (4.17)
	Complex partial seizure, 1 (4.17)
	Myoclonic encephalopathy, 1 (4.17)
	Lennox–Gastaut syndrome, 1 (4.17)
	Aicardi syndrome, 1 (4.17)
	Unknown, 3 (12.5)
Median; IQR; age at epilepsy onset	7 months (36 months)
Median; IQR; age at cKD initiation	24 months (28.5 months)
Median, IQR; cKD duration	21 months (72 months)
Mode IQR number of AEDs	2 drugs (1–5 drugs)
Type of cKD (*n*, %)	cKD 3:1, 15 (62.5)
	cKD 4:1, 9 (37.5)
Administration route	Oral, 19 (79.17)
	Enteral, 5 (20.83)
Overall drop-out rate at 6 months	14 (58.33%)

AED: antiepileptic drugs, cKD: classic ketogenic diet.

**Table 2 nutrients-14-01744-t002:** Biochemical data of study patients on cKD at baseline and after 6 months (*data recorded only 10 patients because of the 14 patients drop out).

Variable	Baseline (Median ± IQR)	6 Months (Median ± IQR)
Total cholesterol (mmol/L)	3.45 ± 1.30	3.50 ± 2.60
Triglycerides (mmol/L)	0.80 ± 0.05	1.76 ± 1.77
ALP levels (U/L)	214.50 ± 119.75	173.00 ± 118.00
ALT levels (U/L)	15.50 ± 10.50	18.00 ± 5.00
GGT levels (U/L)	21.00 ± 8.25	13.00 ± 12.75
Albumin (g/L)	42.00 ± 9.00	44.00 ± 3.00

ALP: alkaline phosphatase, ALT: alanine transaminase, GGT: gamma-glutamyl transferase, cKD: classic ketogenic diet; IQR: interquartile range.

**Table 3 nutrients-14-01744-t003:** Generalized estimating equation modeling.

	Pre–Post	Time × cKD Type Interaction Modeling		
	Modeling			
Variable	MD_time_	MD_time_	MD_type_	MD_timextype_
	*p*-Value	*p*-Value	*p*-Value	*p*-Value
	95% CI	95% CI	95% CI	95% CI
Total cholesterol (mmol/L)	0.210	0.143	−0.290	0.224
	0.75	0.86	0.5657	0.8767
	[−1.09; 1.51]	[−1.40; 1.690]	[−1.28; 0.701]	[−2.60; 3.051]
Triglycerides (mmol/L)	**1.4661**	**1.4571**	0.1571	0.0429
	**0.002**	**0.010**	0.096	0.940
	**[0.507; 2.425]**	**[0.342; 2.573]**	[−0.028; 0.342]	[−1.073; 1.158]
ALP levels (U/L)	−26.4	−1.2	20.3	−93.0
	0.43	0.98	0.71	0.13
	[−92.4; 39.6]	[−82.2; 79.8]	[−87.1; 127.7]	[−213.3; 27.2]
ALT levels (U/L)	6.29	8.10	2.55	−6.35
	0.15	0.17	0.53	0.35
	[−2.34; 14.9]	[−3.60; 19.80]	[−5.39; 10.49]	[−19.71; 7.01]
GGT levels (U/L)	−2.06	−8.2	−20.4	21.4
	0.895	0.701	0.198	0.398
	[−32.8; 28.7]	[−50.2; 33.8]	[−51.6; 10.7]	[−28.2; 70.9]
Albumin (g/L)	**4.30**	2.50	−6.90	5.30
	**0.034**	0.187	0.096	0.234
	**[0.32; 8.28]**	[−1.21; 6.21]	[−15.04; 1.24]	[−3.43; 14.03]

ALP: alkaline phosphatase, ALT: alanine transaminase, CI: confidence interval, GGT: gamma-glutamyl transferase, cKD: classic ketogenic diet; MD: mean difference, SD: standard deviation. Significant values (*p* < 0.05) are in bold.

**Table 4 nutrients-14-01744-t004:** Change in growth of study patients after 12 months on cKD (data collected only in 6 patients, because of drop-out rate).

Variable	Baseline (Median ± IQR)	12 Months (Median ± IQR)
Weight z-score	−0.05 ± 2.00	−0.48 ± 1.76
Height z-score	−0.46 ± 3.29	−0.76 ± 1.92
BMI or (weight-for-length) z-score	0.22 ± 1.98	0.43 ± 1.04

BMI: body mass index, IQR: interquartile range.

## Data Availability

Not applicable.

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
