# Peer review of "Effects of Classic Ketogenic Diet in Children with Refractory Epilepsy: A Retrospective Cohort Study in Kingdom of Bahrain"

_nutrients, 2022, doi:10.3390/nu14091744_

Round 1

Reviewer 1 Report

The authors describe an open label study of the use of the ketogenic diet in 24 children with refractory epilepsy. There are now several studies, including large randomized controlled trials on the use of the ketogenic diet in children with refractory epilepsy, But I don't know what new knowledge will be added to the published literature. I hope authors can give some arguments for it.

Author Response

ANSWERS OF THE AUTHORS TO REVIEWERS

We thank the Reviewers for their appreciation, comments and suggestions. Here added you will find our answers with indications of the changes of the text.

Reviewer 1

Q: The authors describe an open label study of the use of the ketogenic diet in 24 children with refractory epilepsy. There are now several studies, including large randomized controlled trials on the use of the ketogenic diet in children with refractory epilepsy, But I don't know what new knowledge will be added to the published literature. I hope authors can give some arguments for it.

A: Thanks a lot for your positive comments. Regarding question mark on “added value on the published literature”, we shed in light that this study is the first study conducted in Kingdom of Bahrain and in the whole Middle East on cKD treatment for childhood drug-resistant epilepsy which makes it a valuable finding in the region. However, like other studies, it has some limitations including its retrospective nature and the small sample size. Also, it was not possible to exclude the effect of AEDs on liver enzymes and to evaluate the ketonemia levels. It is highly recommended to conduct future research in middle eastern popolation that include larger sample size and prospective study design to assess the nutritional and clinical effect of cKD in short- and long-term. Also, other benefits of KD in terms of quality of life can be investigated in patients with epilepsy. In addition, other less restrictive types of cKD including MCT oil KD, modified Atkin and low-glycaemic index can be included in the upcoming studies.

Reviewer 2 Report

This is an interesting study on ketogenic diet efficacy in a group of children with drug-resistant epilepsy from Bahrain.  Although such studies have already been carried out, the presented results are interesting in the context of the use of a ketogenic diet in a culture where the daily diet is mainly based on carbohydrates, which may explain the fact that many of the studied children gave up the diet. 

The study is generally well designed and the manuscript properly drafted. 

I have a question according to serum beta hydroxybutyrate levels. Can the Authors provide such data and correlate them with KD efficacy?

Another matter that need clarification is concomitant medication.  The Authors should specify them in Methods section. Did KD influenced the medication number/doses during the observation period?

Minor comments:

Table 1 - median number of AEDs - 2.5 drugs? Please check

Page 5 - line 206 - Rett syndrome, line 208 Lennox-Gastaut syndrome

Author Response

Reviewer 2

Q: This is an interesting study on ketogenic diet efficacy in a group of children with drug-resistant epilepsy from Bahrain.  Although such studies have already been carried out, the presented results are interesting in the context of the use of a ketogenic diet in a culture where the daily diet is mainly based on carbohydrates, which may explain the fact that many of the studied children gave up the diet. The study is generally well-design and the manuscript properly drafted. 

A: Thanks a lot for your positive comments.

Q: I have a question according to serum beta hydroxybutyrate levels. Can the Authors provide such data and correlate them with KD efficacy?

A: This study collected data only Total cholesterol, Triglycerides, ALP, ALT, GGT levels and Albumin and correlated them with the KD efficacy.

Regarding the hydroxybutyrate levels were not collected during this study as a part of the study protocol”, but all patients were asked to monitor them, but not to send them to the Medical Team and therefore it was not possible to collect them for analysis because of many missing data

Q: Another matter that need clarification is concomitant medication.  The Authors should specify them in Methods section. Did KD influenced the medication number/doses during the observation period?

A: in order to give a more representative number of medical drugs, we replaced the median with the mode. The mode is 2. A pre-analysis showed that the number of medical drugs AEDs was not a factor associated with KD. Only the “duration” of the KD was associated positively but weak (please look the network weighted analysis, we stated into the results this important factor). In addition, we specified in methods the AEDs drugs included into this protocol of study clobazam, phb, oxcabazepine, levetiracetam, topiramate, phenobarbiton, clonazepam, sabril, vigabatrin, rivotrel, clonazepam

Q: Table 1 - median number of AEDs - 2.5 drugs? Please check

A: replaced with mode.

Q: Page 5 - line 206 - Rett syndrome, line 208 Lennox-Gastaut syndrome

A: done

Reviewer 3 Report

  1. Please structure the abstract
  2. provide study hypothesis and state the primary and secondary objectives clearly
  3. shorten the introduction and please write only about the relevance of KD in refractive epilepsies, and briefly about the types of diet used
  4. each sentence does not need a reference, please correct
  5. some references are very old, for eg ref no 4 is almost 10y old
  6. define epilepsy by ILAE, not by some other author (or your team)
  7. the main problem is the study design, the authors state that it is an open label study but it actually seems a retrospective study
  8. please provide the 'standardized study protocol' mentuoned

Author Response

Reviewer 3

Q: Please structure the abstract

A: done

Q: Provide study hypothesis and state the primary and secondary objectives clearly shorten the introduction and please write only about the relevance of KD in refractive epilepsies, and briefly about the types of diet used each sentence does not need a reference, please correct

A:  We included the study hypothesis and we clearly stated the primary and secondary objectives

Q: some references are very old, for eg ref no 4 is almost 10y old

A: we replaced the very old ref as you suggested.

We replaced ref 4 with updated data:

  • Wirrell EC, Grossardt BR, Wong-Kisiel LC, Nickels KC. Incidence and classification of new-onset epilepsy and epilepsy syndromes in children in Olmsted County, Minnesota from 1980 to 2004: a population-based study. Epilepsy research. 2011 Jun 1;95(1-2):110-8.

Q: Define epilepsy by ILAE, not by some other author (or your team)

A: we included a definition of ILAE based on an appropriate reference.

  • Fisher, R.S., Acevedo, C., Arzimanoglou, A., Bogacz, A., Cross, J.H., Elger, C.E., Engel Jr, J., Forsgren, L., French, J.A., Glynn, M. and Hesdorffer, D.C., 2014. ILAE official report: a practical clinical definition of epilepsy. Epilepsia, 55(4), pp.475-482.

Q: the main problem is the study design, the authors state that it is an open label study but it actually seems a retrospective study

A: You are totally right. We consulted with our statistician team and the correct study is a “Retrospective Cohort Study”. Thanks a lot for sharing this very constructive criticism and provided to fix this issue.

Q: please provide the 'standardized study protocol' mentioned

A: our standardized dietetic protocol follows the “Recommendations of the International Ketogenic Diet Study Group”, so we added the ref: Kossoff, E.H.; Zupec-Kania, B.A.; Auvin, S.; Ballaban-Gil, K.R.; Christina Bergqvist, A.G.; Blackford, R.; Buchhalter, J.R.; Caraballo, R.H.; Cross, J.H.; Dahlin, M.G.; et al. Optimal clinical management of children receiving dietary therapies for epilepsy: Updated recommendations of the International Ketogenic Diet Study Group. Epilepsia Open 2018, 3, 192, doi:10.1002/EPI4.12225.

Round 2

Reviewer 2 Report

The  manuscript has been improved according to my comments.

Author Response

Thank you for your feedback!

Reviewer 3 Report

I have no more corrections

Author Response

Thank you for your feedback!